## RESEARCH ARTICLE

# Bacterial killing assays in ecoimmunology require cross-validation by agreement statistics

Mariah N. Painter*, Rylee C. Conklin, Paige Stephens, Courtney Davis, Maxwell Soell and Franziska C. Sandmeier

## ABSTRACT

Over the last 15 years, an optical density (OD-based) technique to quantify bacterial killing assays (BKAs) has been steadily gaining in popularity. This technique uses spectrophotometry to quantify bacterial growth, rather than the colony counts (CFUs) used previously, and reduces the time, resources, and variability inherent to the assay. However, we argue that the OD-based method relies on assumptions that are not true of all immune components, such as leukocytes, and that methods may not be interchangeable. We performed a targeted literature review focused on the methodology of BKAs across vertebrate taxa in ecoimmunology. We then compared the CFU and OD-based methods using leukocytes isolated from Mojave desert tortoises (*Gopherus agassizii*) and analyzed the quantification method and bactericidal ability using correlation and agreement statistics. Our results suggest poor agreement between techniques, and that immunological processes in cell-based BKAs are likely changing the optical properties of the cultures.

KEY WORDS: Bacterial killing assay, Microbiocidal assay, Ecoimmunology, Reptiles

## INTRODUCTION

The bacterial killing assay (BKA) is a simple, functional assay that assesses the immediate ability of the constitutive immune system to clear pathogens *in vitro* (Tieleman et al., 2005; Millet et al., 2007; Demas et al., 2011). This assay is performed by mixing plasma or whole blood from vertebrate animals with a known concentration of bacteria, incubating for a brief period, and then plating the samples on nutrient agar. Following overnight incubation of agar plates (or a time period similar to mid-log phase growth), bacterial colonies (colony-forming units, CFUs) are counted, then compared against control plates prepared with just media and bacteria to determine the proportion of bacteria killed (Fig. 1A; Millet et al., 2007; Demas et al., 2011). It is a highly adaptable assay that can answer a variety of questions related to the constitutive immune system, depending on the use of immune components (whole blood, isolated white blood cells, plasma, serum) and microbial species included in the assay (Millet et al., 2007; Baker et al., 2019a,b). The assay is especially useful for studying the immune system of a variety of vertebrate species, as it

Colorado State University-Pueblo, Pueblo, CO 81001, USA.

*Author for correspondence (mn.painter@outlook.com)

 M.N.P., 0009-0009-4500-3743; F.C.S., 0000-0002-7043-5902

does not require species-specific reagents (Millet et al., 2007; Tieleman et al., 2005; Demas et al., 2011).

As BKAs became more widely adopted, Liebl and Martin (2009) developed an optical-density (OD)-based method using a Nanodrop spectrophotometer that requires a much smaller volume, fewer resources, and less time. This method was then expanded upon by French and Neuman-Lee (2012) for use with a microplate reader. Both methods are based on quantifying turbidity or absorbance of the sample via spectrophotometry instead of counting CFUs, with only superficial differences (Fig. 1B). Briefly, the assay challenges whole blood or plasma with a known concentration of bacteria and incubates the sample for 30-60 min, tryptic soy broth (TSB) is added to the samples before sample ODs are quantified on either a Nanodrop or microplate reader – this is the baseline value. To quantify microbial growth (an estimate of bacterial viability after the initial incubation), the OD of the sample is quantified again after incubation to reach mid-log phase. The difference in OD readings is used as a proxy for counting plated CFUs (Fig. 1). Both Liebl and Martin (2009) and French and Neuman-Lee (2012) used correlations or regression analyses to show significant, positive relationships between CFU- and OD-based techniques for plasma-based BKAs (Liebl and Martin, 2009; French and Neuman-Lee, 2012). Neither methodological paper performed a cross validation between the two methods using whole blood, although both report that their methods will work with the addition of blood cells (Liebl and Martin, 2009; French and Neuman-Lee, 2012).

However, four main assumptions are embedded in the use of OD-based over the CFU-based technique. (1) The OD-based method assumes that the measured OD primarily quantifies absorbance due to the density of bacteria in solution. While this seems like a reasonable assumption, light is also scattered and different wavelengths will react differently with samples based on many factors such as particle/cell size, density, bacterial cell wall structures, and production of by-products, which can lead to inaccurate growth estimates even in simple bacterial cultures (Dmitiriev et al., 2004; Myers et al., 2013; McBirney et al., 2016). Included in this assumption is that the killing mechanisms themselves will not change the OD of the sample. For example, immune mechanisms such as complement and natural antibodies (NAbs) interacting with bacterial cells and potentially creating cell and protein aggregations, may change OD readings (McBirney et al., 2016; Rodriguez and Voyles, 2020). BKAs have used a wide range of wavelengths to quantify microbial growth (e.g. 300-340 nm; Liebl and Martin, 2009; French and Neuman-Lee, 2012; Beck et al., 2017; and 550-640 nm; de Assis et al., 2013; Baker et al., 2019a,b). Depending on the wavelength chosen for OD quantification, bacterial cells, protein aggregates, and immune cells can both scatter and absorb light (Table 4; Schmid, 2001; Meinke et al., 2007). (2) Because this assay is used to quantify bacteria-killing, the spectrophotometric technique assumes that dead bacteria contribute a negligible amount to changes in optical density, which may not be true if the rate of bacterial growth varies in the assay (e.g. due to low starting volumes, nutrient availability in blood,

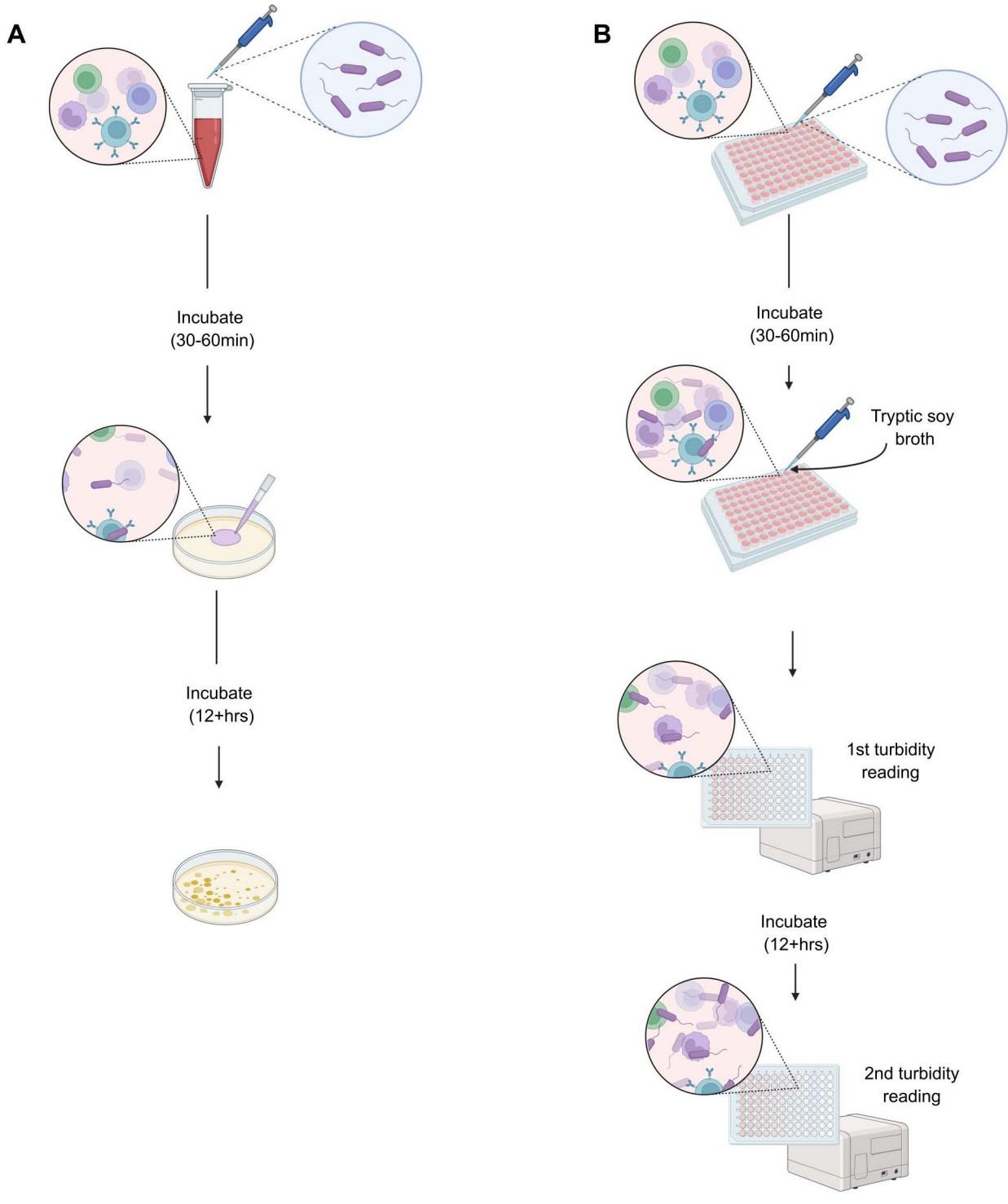

**Fig. 1. Diagram depicting the CFU-based (A) and OD-based (B) methods of the BKA.** For both methods, bactericidal ability is determined by comparing samples against a negative control of microbes grown in media with no immune component. (A) The immune component is prepared in a microcentrifuge or cell culture tube and challenged with a known concentration of bacteria. Samples are then incubated for 30-60 min, after which samples are plated on nutrient agar. Agar plates are then incubated at 37°C for at least 12 h to quantify colonies (Millet et al., 2007). (B) The immune component is prepared in a deep-well plate and challenged with a known concentration of microbes. Samples are then incubated for 30-60 min, after which samples are diluted with nutrient broth. Turbidity is read immediately after diluting, then a second time following a 12+ h incubation. Instead of counting CFUs, bacterial growth is approximated by taking the difference between the two absorbance readings (Liebl and Martin, 2009; French and Neuman-Lee, 2012). Created in BioRender by Painter, M., 2025. https://BioRender.com/lsd96de. This figure was sublicensed under CC-BY 4.0 terms.

etc.; Liebl and Martin, 2009; Stevenson et al., 2016; Hecht et al., 2016). (3) There is an assumption that the immune component being tested will exhaust itself during the initial incubation, and any killing

beyond that will be negligible in liquid culture. This differs from the CFU-based technique, in which only a small sample of bacteria and immune components are spread very thinly on an agar plate

(effectively limiting contact among reagents and halting any further reactions after the initial incubation; Fig. 1A). This assumption has been corroborated for plasma-based BKAs in some species (Adamovicz et al., 2020; Baker et al., 2019b), but leukocytes may be viable for days in media and likely will continue functioning while bacteria are reaching mid-log phase growth in liquid culture (e.g. Slama et al., 2021).

Lastly, (4) the published validations in the literature use correlations among techniques to justify interchangeable use of the CFU- and OD-based BKAs, instead of agreement statistics (Liebl and Martin, 2009; French and Neuman-Lee, 2012). Agreement statistics are widely used in the medical literature and are based on the null hypothesis that the two techniques should result in a 1:1 relationship when they are quantifying the same phenomenon, which should also be the case for BKAs (Ludbrook, 2002; Watson and Petrie, 2010; Ranganathan et al., 2017). Correlations and regressions have the null hypothesis that there is no relationship (Watson and Petrie, 2010; Ranganathan et al., 2017). While significant agreement will result in significant correlations/associations, the reverse often is not true (Watson and Petrie, 2010; Ranganathan et al., 2017). For example, one technique may bias quantification (e.g. over or under-estimate killing; Ludbrook, 2002; Watson and Petrie, 2010) and errors will be compounded in further statistics which are usually the hypotheses being tested in ecoimmunology (e.g. differences in BKA among experimental groups; Demas et al., 2011).

Over about two decades, BKAs has been used widely with plasma (Ruiz et al., 2010; Zimmerman et al., 2010a; Garrido and Mellado, 2015). This is partly due to the fact that plasma can be frozen for extended periods, making it appealing for studies on wild animals (Rubenstein et al., 2008; Gomes et al., 2012; Rynkiewicz et al., 2013). Plasma consists of a number of constitutive immune proteins such as NAbs and complement proteins as well as proteins released from the liver, constitutive cells, and tissues in response to microbial challenge, such as acute phase proteins and antimicrobial peptides (Table 1; Millet et al., 2007; Parham, 2021; Sandmeier, 2024). In wild animals, both constitutive and induced plasma proteins circulate in the blood plasma in varying amounts depending on species, season, infection status, etc., to protect hosts from commonly-encountered microbes (Table 1; Millet et al., 2007; Parham, 2021). Together, these plasma components can kill and disrupt growth of microbes, opsonize microbes to enhance phagocytosis by cells, and modulate immune responses (Table 1; Parham, 2021).

Functions of vertebrate leukocytes are closely tied to plasma proteins, and these cells phagocytize microbes (especially opsonized microbes), produce extracellular nets/traps, release antimicrobial peptides, and modulate a variety of specific and nonspecific immune responses (Table 1; Parham, 2021). Some studies also utilize the BKA with whole blood or leukocytes and plasma (Gervasi et al., 2014; Hartzheim et al., 2023). Using cellular components in a BKA provides a more holistic view of constitutive immune function (Gervasi et al., 2014; Sandmeier, 2024). There is a need in the literature to better understand the constitutive function of ectothermic leukocytes (Ghorai and Priyam, 2018; Zimmerman, 2020), and the BKA could be used to elucidate these functions against a variety of microbes.

Here, we revisit the mechanistic underpinnings of this technique (Fig. 1) to validate its use with blood cells. We first sought to better understand the current state of the literature as it relates to the use of BKAs. We performed a targeted literature review to tally BKA quantification (CFU- versus OD-based), components of the blood that were analyzed (plasma, whole blood/leukocytes, components of both), and taxonomic group(s) of research animals used in each study (fishes, amphibians, reptiles, birds, mammals). In order to determine the validity of using the OD-based BKA using blood cells, we performed a cross validation of the CFU versus OD-based BKA using blood cells with plasma removed from the Mojave desert tortoise (*Gopherus agassizii*; following the methodology of French and Neuman-Lee, 2012). While not the primary focus of this analysis, we also replicated this procedure with plasma for a more direct comparison to the techniques described in Liebl and Martin (2009) and French and Neuman-Lee (2012). We included using comparisons of the OD-based technique at 300 nm (recommended by Liebl and Martin, 2009; French and Neuman-Lee, 2012) and at 600 nm (more commonly used to quantify growth of bacteria; Biesta-Peters et al., 2010; McBirney et al., 2016). We used both correlation (to compare to past studies) and agreement statistics in our comparisons. We predicted that we would find a bias in the literature towards using plasma versus whole blood/leukocytes and using OD-based over CFU-based techniques, given the convenience of those approaches in both the field and laboratory (Millet et al., 2007; Liebl and Martin, 2009). We also predicted that in our study using blood cells, we would find lower agreement than correlation of bactericidal ability between CFU- and OD-based BKAs.

**Table 1. Mechanisms of action of immune cells (leukocytes) and plasma proteins**

| Immune component | Type | Mechanism of action |
|---|---|---|
| Leukocytes | Heterophil | Phagocytosis, release of lysozymes and other antimicrobial peptides, modulation of inflammatory responses, formation of extracellular traps (Montali, 1988; Brinkmann et al., 2004; Parham, 2021; Sandmeier, 2024). |
| | Monocyte | Phagocytosis, release of pro-inflammatory cytokines and antimicrobial peptides, antigen presentation, modulation of constitutive and adaptive responses (Montali, 1988; Origgi, 2007; Parham, 2021). |
| | B1 lymphocyte | Phagocytosis, release of natural antibodies (Baumgarth et al., 2005; Li et al., 2006; Zimmerman et al., 2010b). |
| | Eosinophil | Release of antimicrobial peptides and free radicals, modulate immune responses – especially against parasites (Montali, 1988; Zimmerman, 2016; Parham, 2021). |
| | Basophil | Opsonization, agglutination, modulation of inflammatory responses (Baumgarth et al., 2005; Zimmerman, 2016; Sandmeier, 2024). |
| Plasma proteins | Natural antibodies | Opsonization, membrane attack complex (Parham, 2021; Thau et al., 2023; Sandmeier, 2024). |
| | Acute phase proteins | Complement activation, opsonization, inhibition of microbial growth, immunomodulating actions, tissue repair. Produced by the liver and released in response to infections (Cray, 2012; Parham, 2021). |
| | Antimicrobial peptides | Killing and inhibition of growth of pathogens, immunomodulatory functions, tissue repair. Released from constitutive immune cells and tissues in response to infection (Qiao et al., 2019; Parham, 2021; Santana et al., 2021). |

Both cells and plasma components work together to clear pathogens. For example, cellular phagocytosis is enhanced by opsonization of microbes by plasma components.

## RESULTS

### Literature review

In our literature review, we found 61 ecoimmunological papers on vertebrate species that made use of BKAs (Table 2). By taxa, we found 12 papers focused on birds, 13 on mammals, 25 on reptiles, and 11 on amphibians. Plasma or serum was the immune component used in 48 of the 61 papers (78%), in comparison to whole blood (11/61; 18%). One paper quantifying the immune system of birds used both whole blood and plasma, separately (Matson et al., 2006). One paper quantifying stress physiology in tortoises used plasma and white blood cells, with red blood cells removed (Hartzheim et al., 2023). The 11 papers using whole blood as the immune component were a mix of research based on amphibians, mammals, and birds, and plasma was the only immune component used in all 25 papers studying the reptile immune system. The CFU-based technique quantified bacterial killing ability in 34 studies while the OD-based method was used in 26. One study used a novel BACTEC MGIT method for quantification of viable mycobacteria, which infect macrophages, with whole blood (le Roex et al., 2017). Plasma was the immune component studied in 24 of the 26 papers that used OD to quantify BKA while whole blood was used in just one. (Warfel and Wilcoxen, 2023). The final paper making use of the OD-based method did so with plasma and white blood cells (Hartzheim et al., 2023). For the CFU-based method, plasma was the immune component studied in 24 papers and whole blood in nine papers. Matson et al. (2006) used the CFU-based method in their paper quantifying whole blood and plasma, separately.

### Assay cross-validation

Tortoises showed relatively high variations in differential leukocyte counts (Table 3). We observed values of bactericidal ability below zero and above one in our assay and have left these values unaltered as they represent increased growth due to nutrients in the blood and/or variation in the actual assay.

### Correlation analyses

For cell-based BKAs, the Spearman's rank correlation did not find a statistically significant correlation between CFU counts and OD (300 nm), although there was a weak positive trend (Spearman's $\rho=0.25$; $P=0.11$; Fig. 2A). Similarly, there was not a statistically significant correlation of cell-based bactericidal ability between CFU and OD-based BKAs (300 nm; Spearman's $\rho=0.0244$, $P=0.87$; Fig. 2B). For plasma-based BKAs, there was no significant association between CFUs counts and OD (300 nm; Spearman's $\rho=-0.2451$; $P=0.3430$). There was no significant association of bactericidal ability between CFU and OD-based techniques (300 nm: Spearman's $\rho=0.0368$, $P=0.8886$; Fig. 3A,B).

### Agreement statistics

Cell-based BKAs

The paired $t$-test found a significant difference between the CFU and OD-based bactericidal ability at 300 nm (t=8.64, d.f.=41,

$P<0.0001$) and 600 nm (t=−4.4812, d.f.=26, $P=0.0001$). A Bland-Altman analysis demonstrated a bias of +0.62 (95% CI: 0.477, 0.768), with an upper limit of agreement (ULoA) of 1.54 (95% CI: 1.28, 1.78) and a lower limit of agreement (LLoA) of −0.29 (95% CI: −0.54, −0.042) in the OD-based method at 300 nm when compared to the CFU-based method (Fig. 2C). When comparing the CFU-based method to the OD-based method at 600 nm, the Bland-Altman analysis found that the OD-based method had a bias of +0.47 (95% CI: 0.25, 0.68) with an ULoA of 1.54 (95% CI: 1.17, 1.91) and a LLoA of −0.60 (95% CI: −0.97, −0.22) (Fig. 2D). Lin's concordance analysis indicated no significant agreement of bactericidal ability between the CFU and OD-based methods at either 300 nm ($\rho_c=0.03$) or 600 nm ($\rho_c=-0.03$).

We also performed agreement statistics comparing just the OD-based method of bactericidal ability at 300 nm and 600 nm. The paired $t$-test showed that there was a significant difference in calculated bactericidal ability (t=2.48, d.f.=26, $P=0.02$). The Bland-Altman analysis indicated that the OD-based method read at 300 nm had a bias of +0.20 (95% CI: 0.03, 0.37) with an ULoA of 1.03 (95% CI: 0.74, 1.32) and a LLoA of −0.63 (95% CI: −0.91, −0.33) when compared to the OD-based method read at 600 nm (Fig. 2E). Lin's concordance correlation analysis indicated moderate agreement between the OD-based method read at 300 nm and the OD-based method read at 600 nm ($\rho_c=0.42$).

Plasma-based BKAs

The paired $t$-test found no significant difference between the CFU and OD-based bactericidal ability at 300 nm (t=0.2677, d.f.=16, $P=0.7924$) but found a significant difference at 600 nm (t=2.1298, d.f.=16, $P=0.0491$). A Bland-Altman analysis demonstrated a bias of +0.11 (95% CI: −0.0005, 0.222) when comparing the CFU-based method and the OD-based method at 600 nm, with an ULoA of 0.534 (95% CI: 0.34, 0.73) and a LLoA of −0.311 (95% CI: −0.504, −0.1104). When comparing the CFU-based method to the OD-based method at 300 nm, this analysis revealed a bias of +0.013 (95% CI: −0.09, 0.116) with an ULoA of 0.407 (95% CI: 0.229, 0.593) and a LLoA of −0.385 (95% CI: −0.566 to −0.202). Lin's concordance analysis indicated no significant agreement of bactericidal ability between the CFU and OD-based methods at either 300 nm ($\rho_c=0.061$) or 600 nm ($\rho_c=-0.011$).

Similar to the blood cell-based assay, we also compared just the OD-based method of bactericidal ability at 300 nm and 600 nm using agreement statistics. The paired $t$-test demonstrated significant difference in bactericidal ability (t=3.924, d.f.=16, $P$-value=0.0012). The Bland-Altman analysis indicated that the OD-based method at 300 nm had a bias of +0.098 (95% CI: 0.045, 0.151) with an ULoA of 0.3 (95% CI: 0.207, 0.393) and a LLoA of −0.104 (95% CI: −0.196, −0.0115). Lin's concordance analysis did indicate fair agreement between the two wavelengths ($\rho_c=0.73$).

## DISCUSSION

### Overview

We found that there is a heavy bias towards using plasma in the BKA as opposed to blood. We did not find support for our second hypothesis that there would be a bias towards OD-based methods of the BKA as opposed to CFU-based methods and instead found that there were slightly more papers that used the CFU-based method (Table 2). Overall, the quantification of cellular bactericidal activity lags much behind that of plasma-based bactericidal activity (Table 2). We did not find support for our hypothesis of a statistically significant correlation or agreement of bactericidal ability of cell-based assays between the CFU- and OD-based methods. All of our agreement

**Table 2. Immune component and methodology used across 61 published ecoimmunological papers**

|  | Plasma | Whole blood | Plasma with white blood cells | Whole blood and plasma, separately | Total |
|---|---|---|---|---|---|
| CFU-based | 24 | 9 | 0 | 1 | 34 |
| OD-based | 24 | 1 | 1 | 0 | 26 |
| BACTEC MGIT | 0 | 1 | 0 | 0 | 1 |
| Total | 48 | 11 | 1 | 1 | 61 |

**Table 3. Average tortoise leukocyte counts (± st. dev.) across season**

| Season | Heterophils | Basophils | Lymphocytes | Monocytes | Eosinophils |
|---|---|---|---|---|---|
| Spring | 20.06 (±12.87) | 15.3 (±11.28) | 45.76 (±14.40) | 9.5 (±5.50) | 9.4 (±5.89) |
| Summer | 14 (±8.21) | 17.67 (±14.30) | 42.9 (±14.43) | 11.55 (±5.44) | 13.83 (±10.33) |
| Fall | 25.57 (±15.04) | 20.07 (±10.63) | 31.21 (±13.00) | 10.85 (±4.75) | 12.28 (±7.91) |

statistics suggest that in cell-based BKAs, the OD-based BKAs technique systematically overestimated bactericidal ability compared to the CFU-based method and calculations of bactericidal killing are susceptible to the choice of different wavelengths (Fig. 2B,E). We verified that similar results were obtained for a plasma-based assay, although there was not a clear bias to overestimation by OD-based techniques and slightly more agreement (Fig. 3). Below, we suggest several mechanistic explanations for the overestimation of bactericidal activities and ways in which to remedy using BKAs with cells, which may also be applicable to plasma-based BKAs.

### Agreement statistics

Regardless of the focus of the BKA (plasma or cellular components), the OD-based BKA has thus far not been validated via agreement statistics. Given the importance of understanding cell-based bactericidal assays, an obvious hole in the ecoimmunological literature, the overestimation of bactericidal activity by the OD-based BKA warrants a mechanistic understanding (Fig. 2). In addition to testing the appropriate hypothesis, agreement statistics also quantify systematic differences, or biases, among methods (Ludbrook, 2002). Although different in their approach, the agreement statistics we have used in our comparison (paired *t*-test, Bland-Altman analysis, Lin's concordance analysis), show a consensus of no agreement among techniques – OD-based versus CFU-based and between different wavelengths of OD-based approaches.

Given that approximately half of plasma-based BKAs have used the OD-based BKA, some conclusions from BKAs may need to be re-evaluated. Our findings that neither cell-based, nor plasma-based, BKAs have complete agreement between CFU and spectrophotometry also have implications for plasma-based BKAs in the literature. Likely, the cell-based technique compounds the general problems (see below) with light-scatter and changes in OD values because cells are larger and undergo more extreme morphological changes (degranulation/phagocytosis/aggregation) over a longer time period due to activation by bacteria (Fig. 4). While the previous literature on plasma-based BKAs showed significant associations between colony counts and absorbance based on *P*-values, it did not show $R^2$ or correlation values (Spearman's ρ) that suggest agreement between the two techniques (associations of CFUs and absorbance; Liebl and Martin, 2009; ρ=0.45; French and Neuman-Lee, 2012; $R^2$=0.71, with 30% variation around the line of best fit; de Assis et al., 2013; ρ=0.40). For complete agreement, the null hypothesis is a 45° line through the origin, and both an $R^2$ and a ρ of 1 (Ludbrook, 2002; Watson and Petrie, 2010). Additionally, these analyses in the literature did not evaluate bactericidal activity calculated by the two techniques (Liebl and Martin, 2009; French and Neumann-Lee, 2012) – as CFUs versus absorbance is expected to have a tighter association than the

differences in the estimation of bactericidal activity (Fig. 2). In part because the equations for calculating bactericidal ability differ between the two techniques, differences between OD and CFUs will be compounded when calculating bactericidal activity (Eqns 1 and 2).

The CFU-based BKAs have been criticized for producing more variability among replicates than the OD-based technique (Liebl and Martin, 2009). However, increased variability among replicates in OD-based BKAs can be due to testing for viable (versus viable and dead) bacteria and sampling error in pipetting low quantities of bacteria in animals with high bactericidal ability (Fig. 2A). To assess the effects of sampling error, levels of variation should be assessed as a function of killing ability (e.g. Vanysacker et al., 2014). Such patterns have been quantified for qPCR assays that test for levels of pathogen DNA, with low levels of pathogen leading to higher variation among replicates (Vanysacker et al., 2014; Sandmeier et al., 2017). This phenomenon should not lead to problems in research design, unless assays are not well optimized to host and microbial species.

### Overestimation of bactericidal killing with OD-based BKAs

We suggest two main reasons that OD-based BKAs overestimate killing with cells. First is that the reaction continues in the same liquid culture past the 30-60 min incubation (Figs 1,4). Leukocytes can be viable for days in media and capable of killing during the entire timeframe, including the overnight incubation (Slama et al., 2022; Hartzheim et al., 2023; Nelson, 2025). While heterophils react very quickly, monocytes are long-lived and B1 lymphocytes in non-mammalian and non-avian vertebrates respond much more slowly and may not demonstrate peak antimicrobial capacity until well after that initial incubation period (Parham, 2021; Slama et al., 2022; Nelson, 2025). In contrast, on solid media, the reactions among immune cells and bacteria are largely halted by thinly spreading samples in a film over solid nutrient agar so that cells are no longer in contact with each other (Figs 1,4).

Second, changes in the immune cells themselves may alter OD readings (Table 1). In the equation used by ecoimmunologists to quantify killing via absorbance (Eqn 2), it is assumed that any effect of immune components (e.g. cells as well as plasma proteins) is negligible because the initial readings are subtracted from the final readings (e.g. Liebl and Martin, 2009; French and Neuman-Lee, 2012). This assumption would be correct if immune components were static, when in fact cells, proteins, and microbes change structure, shape, and size as they react, affecting OD readings (Table 4). For example, heterophils/neutrophils are expected to react very quickly in BKAs, leading to degranulation, the release of extracellular traps, and phagocytosis that affect the optical properties of both the immune cells and the surrounding media (Table 4; Mourant et al., 2000; Nelson, 2025). Similarly, early inflammatory processes by both

**Table 4. Optical properties of blood components. Scattering spectra is included where that data is available**

| Component | Absorbance spectra | Scattering spectra |
|---|---|---|
| Red blood cells | 250-500 nm (Bosschaart et al., 2014) | <1000 nm (Bosschaart et al., 2014). |
| Leukocytes | 200-290 nm, 950-1500 nm (Terent'yeva et al., 2017; Liu et al., 2006) | |
| Plasma | 350-550 nm (Meinke et al., 2007) | 350-550 nm (Meinke et al., 2007) |
| Proteins | 180-300 nm (Schmid, 2001) | |

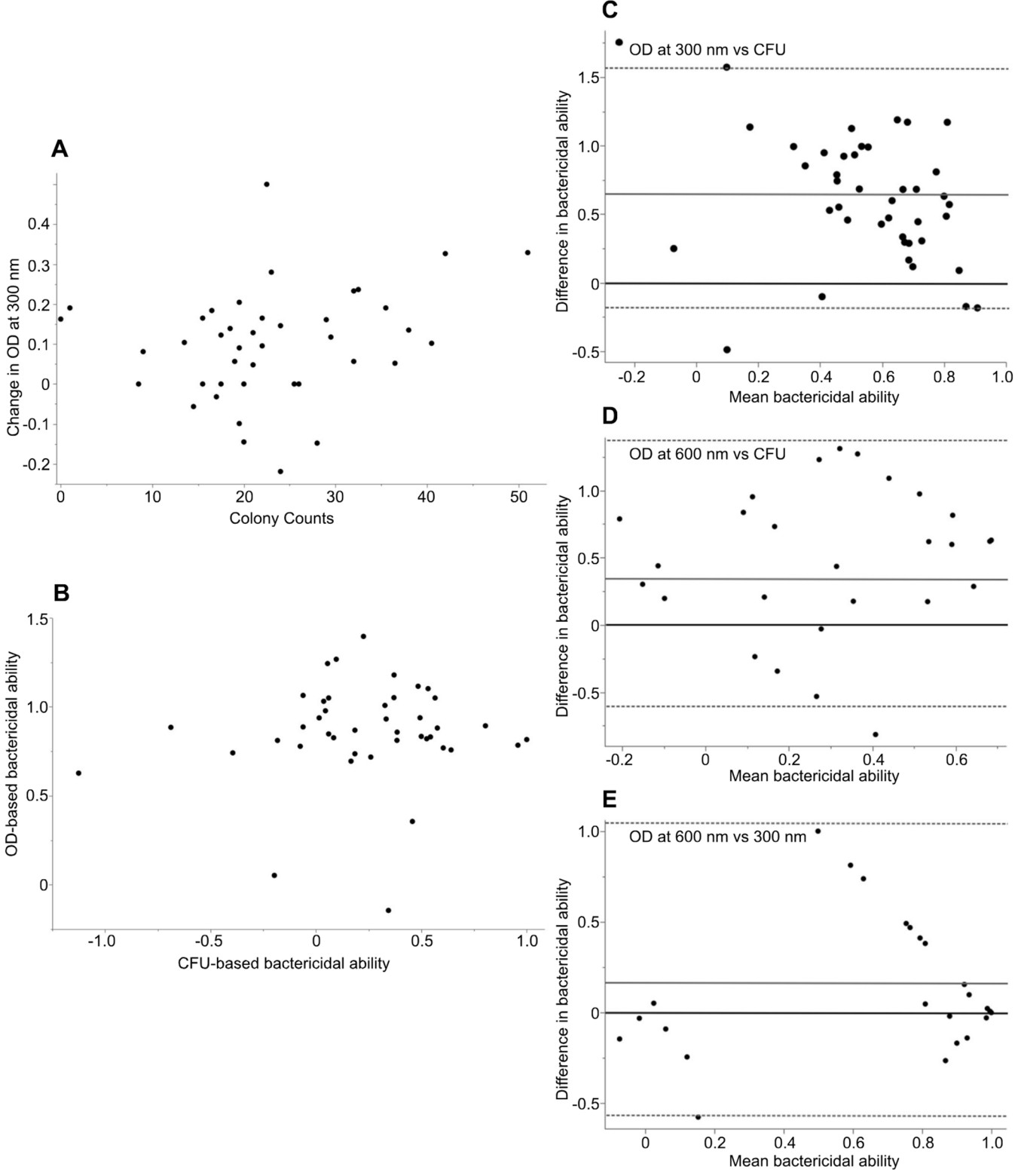

**Fig. 2. Graphs depicting the comparison of colony counts and absorbance and bactericidal ability between the OD and CFU-based methods of cell-based BKAs.** (A) A scatter plot comparing colony counts (CFUs) versus absorbance read at 300 nm for our cell-based BKA run at 35.5°C (*n*=42) (Spearman's ρ=0.25, *P*=0.11). (B) A scatter plot comparing bactericidal ability as calculated by CFUs versus absorbance (*n*=42) (Spearman's ρ=0.024, *P*=0.87). (C) A Bland-Altman plot comparing the bactericidal ability of leukocytes measured via CFU- and OD-based techniques at 300 nm (*n*=42). A solid black line denotes 0. There was a bias of +0.62 for the OD-based method (solid grey line), with limits of agreement of −0.29 to 1.54 (dashed lines). (D) A Bland-Altman plot comparing the bactericidal ability of leukocytes measured via the CFU and OD-based methods at 600 nm (*n*=27). A solid black line denotes 0. There was a bias of +0.47 for the OD-based method (solid grey line), with limits of agreement of −0.60 to 1.54 (dashed line). (E) A Bland-Altman plot comparing the bactericidal ability of leukocytes measured by the OD-based method at 300 and 600 nm (*n*=27). A solid black line denotes 0. There was a bias of +0.20 for the OD-based method (solid grey line), with limits of agreement of −0.63 to 1.03 (dashed lines).

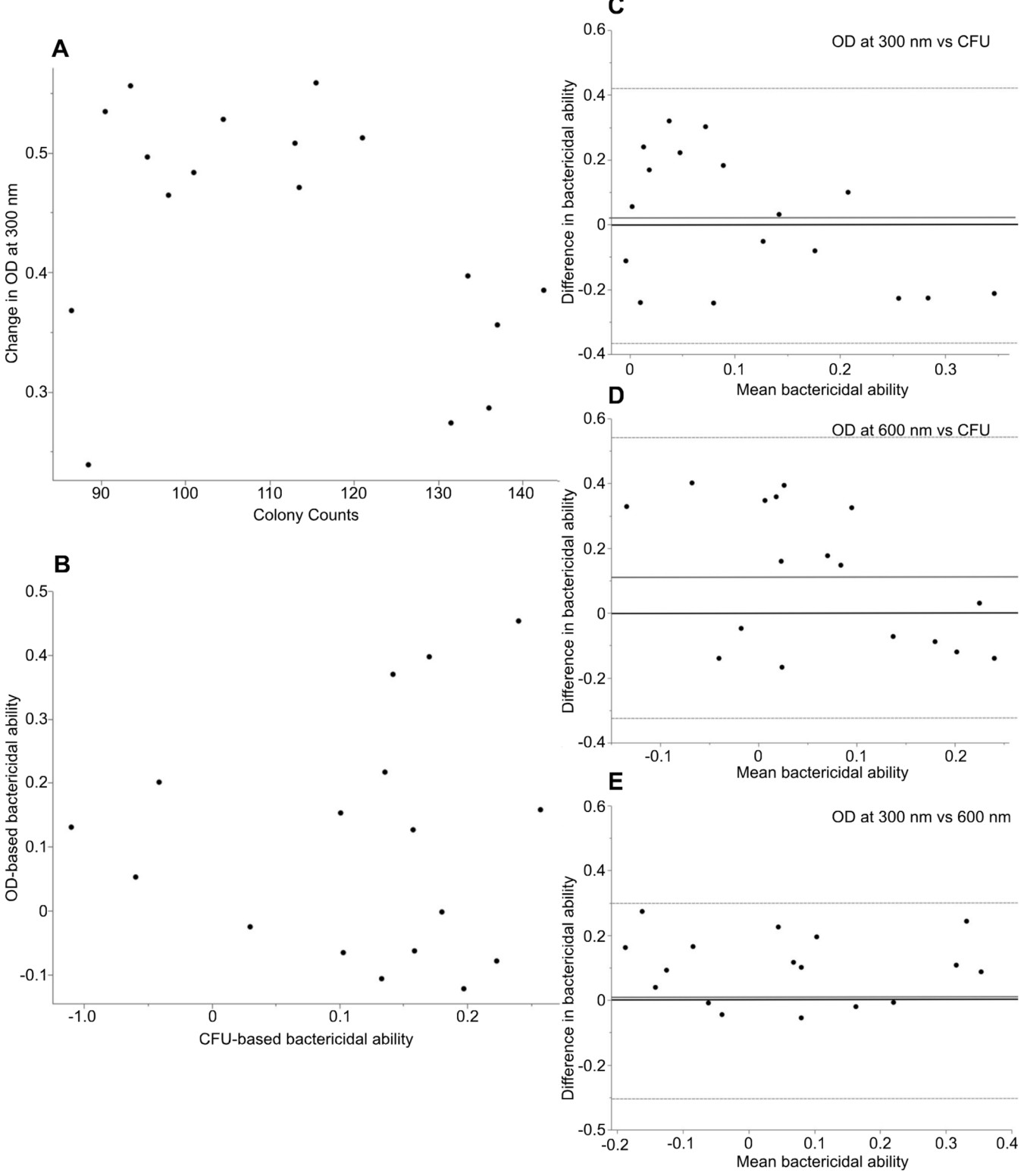

**Fig. 3. Graphs depicting the comparison of bactericidal ability between OD- and CFU-based methods of plasma-based BKAs.** (A) A scatter plot comparing colony counts (CFUs) versus absorbance read at 300 nm for our cell-based BKA run at 35.5°C (*n*=17) (Spearman's ρ=−0.2451, *P*=0.3430). (B) A scatter plot depicting bactericidal ability as calculated by CFUs versus absorbance at 300 nm (*n*=17) (Spearman's ρ=0.0368, *P*=0.8886). (C) A Bland-Altman plot comparing the bactericidal ability of leukocytes measured via CFU and OD-based techniques at 300 nm (*n*=17). A solid black line denotes 0. There was a bias of +0.013 for the OD-based method (solid grey line), with limits of agreement of −0.385 to 0.411 (dashed lines). (D) A Bland-Altman plot comparing the bactericidal ability of leukocytes measured via the CFU and OD-based methods at 600 nm (*n*=17). A solid black line denotes 0. There was a bias of +0.11 for the OD-based method (solid grey line), with limits of agreement of −0.311 to 0.534 (dashed line). (E) A Bland-Altman plot comparing the bactericidal ability of leukocytes measured by the OD-based method at 300 and 600 nm (*n*=17). A solid black line denotes 0. There was a bias of +0.0982 for the OD-based method (solid grey line), with limits of agreement of −0.104 to 0.3 (dashed lines).

0.5-1hr

12hrs

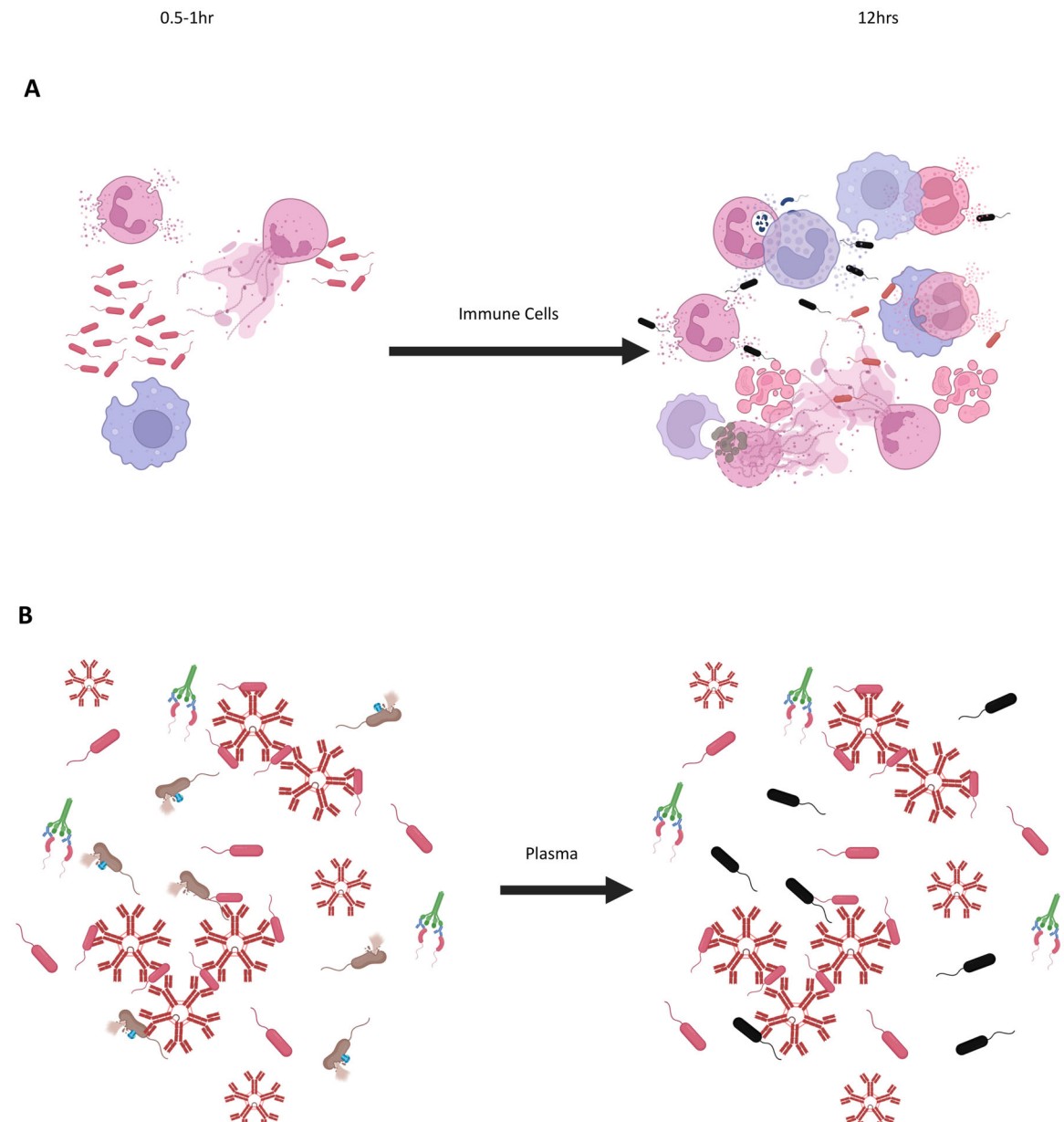

**Fig. 4. Illustration of the mechanisms of killing present *in vitro* within the first hour of an immune response and after 12 h for (A) isolated immune cells and (B) isolated plasma.** In both diagrams, live bacteria are depicted in grey and dead bacteria are depicted in black. (A) In assays using isolated leukocytes, heterophils and, to a lesser extent, monocytes will be the first cell types to respond when challenged with bacteria. More cell types will be recruited as time progresses and killing will continue throughout a 12 h incubation period. (B) Plasma proteins are expected to be expended after 0.5-1 h and any killing beyond that is likely negligible. If both plasma and blood cells are combined, we expect even higher rates of continued cellular killing, due to opsonization. Created in BioRender by Painter, M., 2025. https://BioRender.com/l6mibpz. This figure was sublicensed under CC-BY 4.0 terms.

heterophils/neutrophils and monocytes may cause aggregation of cells in the liquid culture that could impact absorbance readings (Parham, 2021; Nelson, 2025). Apoptosis of leukocytes over longer time frames will also alter OD readings (Liu et al., 2001; Gault and Lefaix, 2003).

In support of the idea that immune cells are changing OD values is the estimation of more than 100% killing in our data, only for the cell-based assays (Fig. 2), as well as by others that have used OD-based BKAs with leukocytes and plasma (e.g. Hartzheim et al., 2023; Terry et al., 2023). Values above 100% killing are only possible if the OD readings of experimental samples actually decrease as the samples are incubated overnight (Fig. 1; Eqn 2). This makes relatively little sense if the OD only quantifies absorbance of bacteria, especially since the small amount of bacteria initially

added to the assay do not significantly change OD readings of the media ( pers. observation). Instead, these calculations suggest that all of the bacteria had been killed, and the background absorbance is lost as well. In our interpretation, the only logical reason for this pattern are different levels of light scattering and absorbance as leukocytes are lysed and/or as blood cells/fragments of cells/ proteins aggregate and change the sizes of suspended particles (Table 4). This pattern will also be affected by the wavelength used to quantify OD, as is evident in our own analyses (Fig. 2E). While it may be tempting to adjust all these higher values to 1, these mechanisms are affecting every sample in the plate, and setting a limit will distort data and affect the interpretation of the assay. Whether similar patterns exist in assays only based on plasma was

not very clear from our analysis of the literature and were not observed in our data. However, natural antibody and complement proteins have the potential capacity to impact the properties of light attenuation, and through their very actions (zymogen cleavage, aggregation, agglutination of microbes, etc.) could change the shape, size, and density of particles in suspension (Schmid, 2001; Rodriguez and Voyles, 2020; Parham, 2021).

Conversely, when values are below 0% killing, those values indicate that experimental samples have higher OD readings and more growth than the control samples and are not problematic if also observed in the CFU-based BKA (Figs 2B, 3B). Such values quantify increased growth of bacteria, likely due to the many nutrients found in vertebrate blood (Hickman et al., 2024). While some authors adjust these values to zero (Lievesley and Rollinson, 2021; Hartzheim et al., 2023; Terry et al., 2023), we again advise against this strategy as it distorts the data of the whole plate and in fact quantifies the interplay between microbial growth and killing by the immune system (Cressler et al., 2014; Noelker et al., 2024).

### New directions in the use of BKA

In light of several decades of use of BKA assays, we suggest (1) careful re-interpretations of past data, using CFUs or other measures of viable bacterial counts as the gold standard, (2) quantification of changes in optical density of activated immune components (proteins and cells), and (3) exploration of alternative ways to quantify microbial growth without the use of spectrophotometry. First, data to compare CFUs versus spectrophotometry is relatively easy to generate and many researchers may already have unpublished, paired data. These data should be compared with agreement statistics to confirm or reject our conclusion, across species, that there is little agreement in our own data and that of previously published comparisons. Similarly, heteroskedasticity of error values need to be quantified. If spectrophotometry consistently increases all values of cell-based BKAs in a systematic way – without a funnel effect of over- or under-estimating values at the lower and upper ends of bactericidal killing – experimental conclusions in the ecoimmunological literature likely remain valid. However, quantification of error and implications of study design need to be carefully considered in the future.

We suggest several experimental tests of changes in optical properties of immune components, which will help to elucidate mechanistic modes of killing beyond simple BKA values across species and antigenic stimulants (e.g. Millet et al., 2007; Rodriguez and Voyles, 2020). For example, heat-killed bacteria or common antigens (e.g. LPS, ovalbumin, microbeads) stimulate immune pathways without the use of replicated microbes (Demas et al., 2011; Slama et al., 2021). These variably bind to natural antibodies, stimulate the different pathways of complement, and induce phagocytosis in leukocytes (Demas et al., 2011; Parham, 2021). Such antigens should be used across host species and immune components (plasma/serum, isolated leukocytes, whole blood) in spectrophotometric studies to quantify the putative changes in OD in the absence of microbial growth. Such quantifications would allow for re-interpretations of past ecoimmunological studies, while simultaneously quantifying specific mechanisms of immune function against certain antigens.

le Roex et al. (2017) used a Bactec system to quantify live mycobacteria; a microbe which presents challenges in quantifying viability due to its infection of macrophages. Such systems could be used more generally in ecoimmunology, but likely are not widely available to researchers not based at medical institutions. However, flow cytometry and FACs systems are becoming more accessible, and methods could be pioneered for quantifying bacterial cells,

which can be differentiated from protein aggregates and immune cells based on size and complexity (Parham, 2021). Several parameters of such assays would need experimentation and optimization, tailored to incubation times and experimental design. Possible targets for optimization could include quantifying viable versus dead bacterial cells by staining, changes in immune proteins (e.g. cleavage, aggregation, opsonization), changes in leukocyte morphology and number (due to lysis), etc. If different staining and gating protocols can accurately quantify opsonized microbes, protein aggregates, changes in cell structure, etc., these new targets of a BKA would greatly increase the mechanistic understanding of constitutive immune function across host species. Additionally, such mechanistic insights would allow for further interpretations of the existing BKA literature and lead to new study questions and experimental designs.

### Conclusion

Here, we showed that BKAs remain an important component of the vertebrate-focused, ecoimmunological literature across taxa and using both CFU- and OD-based quantification techniques. However, adopting this assay to cell-based bactericidal quantifications, showed that validations of the OD-based BKA resulted in a large over-estimation of killing, and suggest revalidations of these spectrophotometric methods across BKAs using appropriate agreement statistics. Finally, we suggest that any inaccuracies due to OD-based calculations actually shed light on mechanisms of immune function and should be explored in greater detail, using both non-replicating antigens and microbes. We hope that this work will lead to increased excitement and creative new research designs of BKAs, in an effort to marry this classic ecoimmunological technique with more mechanistically driven questions of comparative immunology.

### MATERIALS AND METHODS
#### Literature review
We performed a review of the current literature with the intent of quantifying the relative frequency of different types of methodologies used with BKAs in studies of vertebrate ecoimmunology. We searched the following terms in Web of Science: "bacterial killing assay", "bacteria* kill* assay*", "bactericidal assay*", "microbiocidal assay*", and "antimicrobial assay*". We used Web of Science to specifically target the ecoimmunological literature, and we removed literature focused on humans or lab animals as well as veterinary papers focused on identification of specific diseases in domesticated animals. We also reviewed all pertinent references in three review papers, focused on aspects of BKA in ecoimmunology (Demas et al., 2011; Becker et al., 2019; Rodriguez and Voyles, 2020). After collection, we categorized papers by use of different immune components (e.g. whole blood, plasma, leukocytes), techniques to quantify bacteria (counting CFUs, absorbance, other), and taxa (fishes, amphibians, reptiles, birds, mammals).

#### Animal care and sampling
All procedures involving animals were reviewed and approved by the Institutional Animal Care and Use Committee (IACUC) of Colorado State University-Pueblo (IACUC protocol number: 000-000A-038). We used eighteen captive-raised Mojave desert tortoises with no known exposure to disease for this experiment. The tortoises were roughly 35 years of age at the time of sampling and consisted of seven females and 11 males. All tortoises used in this experiment are subjected to regular disease screenings, including physical examination and molecular testing for the presence of *Mycoplasma agassizii*, and have always received negative results. We housed tortoises communally within a greenhouse at Colorado State University-Pueblo and fed them an *ad libitum* diet of grass, alfalfa, and/or leafy green vegetables, supplemented twice weekly with calcium and vitamins. Tortoises were free roaming, had access to natural lighting, and were provided with heat lamps to thermoregulate year-round.

We collected blood samples (0.4-1.0 ml) via subcarapacial venipuncture (Hernandez-Divers et al., 2002) using a heparinized syringe during March (*n*=17), June (*n*=18), and September/October (*n*=13) of 2023 to collect blood cells and during August/September of 2025 to collect plasma (*n*=17). Samples were transferred to heparinized collection tubes and stored on ice for no longer than 1 h. Blood smears were made from whole blood and stained with Wright-Giemsa reagents to quantify differential white blood cell counts, based on 100 leukocytes. Following sample collection, we transferred whole blood to a sterile 2 ml microcentrifuge tube and spun it down for 8 min at 4°C, 300×*g* to remove the plasma. Once plasma was removed, we diluted each blood sample to three times its original volume with cell culture medium (RPMI-1640, GE Lifesciences, Marlborough, MA, USA) supplemented with fetal calf serum and L-glutamine. Due to small blood sample volumes and a preference to conduct BKAs as quickly as possible, we did not quantify cell viability in these particular assays, but heterophils, monocytes, and lymphocytes have been shown to remain viable after identical harvesting techniques in phagocytic assays using fluorescent beads, over longer incubation times (2-4 h; Slama et al., 2022; Nelson, 2025). In fall of 2025, we separated blood plasma and froze it for 1-2 months at −40°C.

### Selection and preparation of bacterial samples

*Escherichia coli* (ATCC #8739) was maintained as an active culture of bacteria throughout each sampling season by incubating in TSB at 37°C. Previous to use in this study, it had been passed through culture as a part of other ecoimmunological studies. At the same time each morning, we transferred 1 ml of the previous days' culture to 2 ml of fresh TSB and maintained the previous culture at 4°C. 24 h prior to sampling, we plated serial dilutions of the active culture ranging from 1:4000 to 1:164,000 on tryptic soy agar (TSA) plates. We maintained dilutions at 4°C for use the next day and incubated plates at 37°C for 12+ h. We counted colonies the following morning prior to sampling and chose dilutions that allowed us to add roughly 5 µl of bacteria to 145 µl of blood dilution.

### BKA

For cell-based assays, we added *E. coli* in TSB to each sample at a concentration of 200 CFUs per 150 µl of blood dilution and mixed samples gently by pipetting. For plasma-based assays, we added 20 µl of plasma to 500 CFUs of *E. coli*, diluted in Luria Broth (LB), in a total volume of 200 µl of solution (*sensu* Sandmeier et al., 2023). For both assays, we prepared a positive control by adding the same amount of *E. coli* in the respective cell media used for each BKA to the same volume of supplemented cell media without blood cells or plasma. We incubated all samples for 30 min at 35.5°C (MyTemp Mini CO$_2$ incubators, Midland Scientific, Aurora, CO, USA). During the fall season, we incubated split samples of the cell-based BKA at 17.5, 26.5, and 35.5°C. These different temperatures were part of a study aimed at quantifying variability among temperatures (Painter, 2025), and data were used incidentally due to the use of absorbances set at both 300 and 600 nm to quantify turbidity in this set of BKAs. This assay and these conditions previously had been optimized for roughly 20-80% killing for this species (Sandmeier et al., 2023; Painter, 2025). Following incubation, we pulled samples from the incubators and placed them on ice to inhibit any further killing during plating.

To conduct the CFU-based BKA, we plated 25 µl of each sample and control cell suspension or 45 µl of each plasma suspension and their respective controls on TSA plates in duplicate. Plates were incubated for 15 h at 37°C and colonies were counted the following morning. The proportion of bacteria killed was calculated by finding the difference in CFUs between the control and sample plates, then dividing that difference by the control CFUs (Eqn 1).

$$Bactericidal\ ability = \frac{(CFU\ Control - CFU\ Sample)}{CFU\ Control} \quad (1)$$

To conduct the OD-based BKAs with both blood cells and plasma, we followed the methodology described in French and Neuman-Lee (2012). Briefly, we mixed each sample and added a 25 µl aliquot of sample (blood-bacteria mixture after 30 min incubation) or control to 125 µl of tryptic soy broth in a sterile round-bottom 96-deep well plate with a cover (Fig. 1B).

Each sample and control were prepared in duplicate. We also prepared a negative control consisting of our 1:3 blood dilution and TSB or plasma with no bacteria on each plate as a negative control. Immediately after adding each sample and control to the deep well plate, we read turbidity at 300 nm (BioTek Synergy LX Multimode reader, Agilent Technologies, Santa Clara, CA, USA). During the fall season, we added an additional 600 nm reading, as this is the wavelength most commonly used in bacteriological studies to quantify growth of *E. coli*. We incubated plates for 15 h at 37°C. We then removed plates from the incubator, vortexed them thoroughly, and read turbidity again at 300 nm (all seasons) and 600 nm (fall only). We determined bacterial growth by calculating the change in OD between the two readings. We then calculated bactericidal ability by comparing the change in OD in the control wells against the change in OD in the sample wells (Eqn 2).

$$Bactericidal\ ability = \frac{(\Delta OD\ Control - \Delta OD\ Sample)}{\Delta OD\ Control} \quad (2)$$

### Statistical analyses

#### BKA

We used a non-parametric Spearman's rank correlation for comparisons, as only bactericidal ability calculated by CFUs was normally distributed (following Liebl and Martin, 2009; French and Neuman-Lee, 2012). We compared CFUs to OD (300 nm) for cell-based and plasma-based BKAs and we compared bactericidal ability determined by CFUs to bactericidal ability determined by OD (300 nm) for both cell-based and plasma-based samples.

In addition to correlation analyses, we ran more appropriate agreement statistics among bactericidal ability calculations by OD (at 300 and 600 nm) and CFU-based methods (paired *t*-tests, a Bland-Altman analyses, and a Lin's concordance analyses; Watson and Petrie, 2010; Ranganathan et al., 2017). A paired *t*-test is a simple way to test against a null hypothesis of no difference between two paired values (e.g. bactericidal ability tested from the same sample via the two methods). A Bland-Altman analysis is a visual depiction of the difference between two methods and includes calculations of the mean of pairs of values (e.g. calculated via two different techniques) and the standard deviations of their differences (e.g. pairs here are bactericidal activity calculated via our two different techniques; Ludbrook, 2002; Watson and Petrie, 2010, Ranganathan et al., 2017). A Bland-Altman plot shows (a) deviation from the null expectation of zero, bias of over or under-estimation of one technique, and the upper and lower levels of agreement (mean±2 s.d. of the difference; Ludbrook, 2002; Watson and Petrie, 2010). If these limits of agreement are acceptable to the research design, one technique can be substituted for the other (Ludbrook, 2002; Watson and Petrie, 2010). Lin's concordance analysis is similar to a Pearson correlation but assesses how far points are from the expected 45° line through the origin that would occur with perfect agreement (Watson and Petrie, 2010). Lin's concordance coefficient is bounded by 1 (perfect agreement) and −1 (no agreement), and a 95% confidence interval of this correlation coefficient can show high repeatability between two measures or techniques (Watson and Petrie, 2010). In this statistical method, values near zero indicate poor agreement, while negative values indicate severe discordance.

The paired *t*-test, Bland-Altman analysis, and Lin's concordance analysis were used to calculate agreement between bactericidal ability using the CFU-based method and the OD-based method at both 300 and 600 nm. We also used these agreement statistics to compare bactericidal ability calculated by OD read at 300 nm and 600 nm from the samples assessed in the fall season. To compare techniques for plasma-based BKAs we repeated these analyses across all categories (CFU-based techniques and the OD-based technique at 300 and 600 nm). Bland-Altman analysis and Lin's concordance analysis were performed using the blandr (v0.60; Datta, 2017) and epiR (v2.0.82; Stevenson and Sergeant, 2025) packages, respectively, in R Statistical Software (v4.42; R Core Team 2024). All other statistical analyses were conducted using JMP® software (SAS institute Inc.).

### Acknowledgements

We would like to thank the Biology Department at Colorado State University of Pueblo for their continued support throughout this project. Materials and Methods/Results in this paper are reproduced from the Master's thesis of Mariah Painter (Colorado State University-Pueblo, 2025).

## Competing interests

The authors declare no competing or financial interests.

## Author contributions

Conceptualizations: F.C.S.; Data curation: M.N.P., R.C.C.; Formal analysis: M.N.P.; Funding acquisition: F.C.S.; Investigation: M.N.P, R.C.C., P.S., C.D., M.S.; Project administration: F.C.S.; Supervision: M.N.P, F.C.S.; Writing – Original Draft: M.N.P.; Writing – Review & editing: F.C.S.

## Funding

This material is based upon work supported by the National Science Foundation under award no. 221866. Open Access funding provided by the National Science Foundation. Deposited in PMC for immediate release.

## Data and resource availability

The data that support the findings of this study are openly available on Dryad at http://doi.org/10.5061/dryad.z8w9ghxtb.

## Peer review history

The peer review history is available online at https://journals.biologists.com/bio/lookup/doi/10.1242/bio.062411.reviewer-comments.pdf

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
