## [Peer Review File · Biology Open]

1 Bacterial killing assays in ecoimmunology require cross-validation by agreement statistics

Rylee C. Conklin, Paige Stephens, Courtney Davis, Maxwell Soell, Franziska C. Sandmeier and Mariah Painter

DOI: 10.1242/bio.062411

Editor: Kendra Greenlee

Review timeline

Original submission:	8 December 2025
Editorial decision:	16 December 2025
First revision received:	17 March 2026
Accepted:	22 March 2026

Original submission

First decision letter

MS ID#: bio.062411

MS Title: 1 Bacterial killing assays in ecoimmunology require cross-validation by agreement statistics

Authors: Mariah Painter; Rylee C. Conklin; Franziska C. Sandmeier

I have now reached a decision on the above manuscript.

The reviewer reports are shown at the bottom of this email.

As you will see, the reviewers raised a number of substantial criticisms that prevent me from accepting the paper at this stage.

They suggest, however, that a revised version might prove acceptable, if you can address their concerns. In particular, I recommend focusing on the clarifications and additional data requested from reviewer 1, along with the comments from reviewer 2 about possible *Mycoplasma*, rationale for not testing plasma, and including a table of data from the literature review. Unless it is feasible, I would not ask for additional experiments as recommended by reviewer 1. If you think that you can deal satisfactorily with the criticisms on revision, I would be pleased to see a revised manuscript.

At this stage, we also ask you to ensure your manuscript complies with our formatting guidelines. Provided you are able to fully address the referees' comments, we are positive about publication of your paper (we accept over 95% of revision submissions) and therefore hope you won't mind any extra work involved in reformatting your manuscript at this point.

Please upload both a 'clean' version of your Word file, along with a highlighted version clearly showing where you have made changes in the revised manuscript. Please avoid using 'Track changes' in Word files as these are lost in PDF conversion.

I should be grateful if you would also provide a point-by-point response detailing how you have dealt with the points raised by the reviewers in the 'Response to Reviewers' box. Please attend to all of the reviewers' comments. If you do not agree with any of their criticisms or suggestions please explain clearly why this is so.

Reviewer 1

Comments for the author

Summary

This manuscript addresses an important methodological question in ecoimmunology: whether OD based bacterial killing assays (BKAs) can be used interchangeably with CFU based approaches, particularly for cell based assays. The study combines a literature review with experimental cross validation and applies agreement statistics, which is a significant improvement over prior correlation based validations. The findings, poor agreement and systematic OD bias, are highly relevant to the field.

However, a few aspects of experimental detail, reproducibility, and statistical rigor require clarification or enhancement before the manuscript can be considered for publication. These revisions will strengthen transparency, interpretability, and the impact of the work.

Major Points

1. Clarify cell preparation and composition: The manuscript refers to "leukocytes," but the methods describe diluted whole blood without RBC removal. Please clarify whether RBCs were present and, if so, provide cell counts. Discuss how nucleated reptilian RBCs may affect OD readings. If feasible, include a subset experiment with RBC reduced or isolated leukocytes to test OD bias mechanisms.
2. Standardize and report cell numbers and viability: Provide leukocyte counts, viability assessments, and any normalization steps. Without standardization, variability may reflect cell abundance rather than method differences.
3. Detail bacterial inoculum and calibration: Specify strain and CFU calibration method. Confirm how the 200 CFU inoculum was verified.
4. Explain and/or stratify wavelength and temperature analyses: OD readings at 600 nm were limited to fall season. It would be helpful to the reader to understand this experimental decision. Raises the question if analysis should be stratified by temperature.
5. Correct figure legends and add sample sizes: Bias values in Figure 2 legends do not match text; please correct and annotate N per comparison.
6. Temper conclusions about plasma based OD BKAs: Current data are cell based; extrapolation to plasma should be framed as a hypothesis or include a brief validation of this claim.

Minor Points

1. Ensure consistent terminology ("cell based BKA" vs "leukocyte based BKA").
2. Add interpretation guidance for Lin's concordance coefficient (values near zero = poor agreement; negative = severe discordance).

Recommendation

The manuscript is conceptually strong and addresses a critical gap in ecoimmunology methods. With the above revisions, it will make a valuable contribution to the field. I encourage resubmission after these changes.

Reviewer 2

Comments for the author

a. Does each figure have the proper controls?

Figures 1 & 3 are diagrams, so not relevant. Figure 2 presents original data and a statistical analysis of this data. The experiments conducted included control reactions. In short, yes.

b. Are experiments performed using appropriate methods that will answer the question (or test the hypothesis or support the observations) posed by the authors? Is the right tool used for the job?

Yes, all methods are thorough and adequate to answer the question/hypothesis posed. Methods, reagents, and equipment are correct - the right tools were used.

c. Were the data analysed using appropriate statistical tests?

Statistical tests are appropriate for the data and software packages used are suitable.

2. Reproducibility

a. Were experiments in each figure performed using adequate number of biological replicates?

Although the total number of samples is not stated anywhere, each plot in Figure 2 has (approx.) 25-40 data points, which is adequate.

b. Is there sufficient raw data to assess the rigor of the analysis?

I assume the link on the final page of the manuscript is to the original data. I am unable to open the link, so I would like the authors to confirm if this is true. If so, then yes, it has been supplied.

c. Does the methods section provide sufficient detail to permit reproducibility?

Yes

3. Completeness

a. Are the author's conclusions supported by the data?

The conclusions are supported by the data and its statistical analysis.

b. Are there any flaws in the experimental design that invalidate the approach taken by the authors?

Although not a flaw in the design, I would be interested to know the authors' response to this: Mojave Desert tortoises commonly carry the bacterium *Mycoplasma agassizii* as a subclinical infection. This pathogen is the primary cause of upper respiratory tract disease, and many individuals are lifelong asymptomatic carriers. I am aware the last author has published studies on this pathogen, so is a known expert. However, there is nothing in the methods indicating the subjects were free of this. I feel it is conceivable that subclinical carriers may have different leukocyte profiles or altered innate immune responses compared to uninfected individuals. Could the authors please comment on this and/or alter the methods section to indicate confidence in the infection status of the subjects used in the study?

c. Are there experiments that have not been performed, but if true would disprove the conclusion? If yes, and if such experiments would be costly or time-consuming to perform, do the authors acknowledge this in a discussion of the limitations?

There are only experiments not performed that would support the conclusion - namely FACS and flow cytometry. The authors acknowledge these, but they are not necessary for this study.

I wondered why the authors did not also test plasma? As they state, most published studies use plasma in the BKA as opposed to blood. So it would seem pertinent to investigate this alongside blood, potentially strengthening the central argument/question through offering a direct comparison. However, it is not obligatory to include, just that it would strengthen the manuscript.

4. Scholarship

a. Do the authors cite and discuss the merits of relevant data that would argue against their conclusion?

Indirectly - by comparing their alternative approach to methods already published (for example, on lines 373-380), which they believe are flawed.

b. Do the authors cite and discuss the merits of relevant data that would support their conclusion?

Yes, in the subsection "New directions in the use of BKA", they refer to several general concepts in immunology that are relevant and support their central hypothesis and their data. Furthermore, on lines 323-326 and 332-339 earlier in the discussion, they identify other studies and/or concepts that are used to support their hypothesis, data and conclusions.

c. For techniques/methods manuscripts, Do the authors cite and discuss the current state of the field and clearly explain how the method improves the field?

Yes, clearly.

This manuscript presents a thoughtful and well-considered idea, principally that fundamental flaws exist with bacterial killing assays that use cells; and use spectrophotometry as the read out. Such that there is a lack of agreement with other bacterial killing assay methods. The authors suggest this may be causing multiple labs to incorrectly analyse and/or report results from these assays. The hypothesis of this study is clear, as is the data and interpretations and conclusion.

Here are a selection of comments for suggestions to improve the manuscript and/or aid clarity:

1. It would be useful for those that will read the paper if the data from the literature review was tabulated with relevant parameters highlighted in specific columns.

2. Whilst this isn't a systematic review, I wonder if the authors also searched other databases such as PubMed or Google Scholar as part of the literature review? Web of Science is generally considered more stringent and selective in its journal inclusion criteria than PubMed.

3. Something I feel should be added to the discussion: one other element that may be relevant and further validate the central point is interindividual variation in the % of different leukocyte populations in blood. I am not aware of how these parameters vary in tortoises (it may not be known), but if we were to use humans as the reference point, neutrophils show a large amount of variation (in %) between individuals.

4. What was the total number of samples tested? This is not mentioned anywhere and should be.

5. Normally, I would expect ethical clearance was granted to acquire tortoises and to perform controlled procedures such as taking blood. This information (e.g. a study code and who gave approval) is missing from the methods section and should be added.

6. Some brief comments referring to specific parts of the manuscript:

Line 116-117: this sentence: "There is a need in the literature to better understand the constitutive function of leukocytes (Sandmeier, 2024a,b)". I agree, however, this is a general point that has

widely been discussed, investigated and commented on. I would therefore feel more comfortable if the references (self-citations) for this point were replaced with others from an "external" source.

Line 158: "briefly transferred to heparinized collection tubes and stored on ice". Please specify for how long they were on ice. Just to be more accurate and eliminate ambiguity. Of course, sample integrity would be compromised if storage at this temperature extended beyond 24h.

Line 163 onwards: A positive control is specified. Would the inclusion of a negative control (blood cells but no bacteria) also be valuable for this?

Some minor typographical errors I noticed:

Line 46: the word "for" is missing before 30.

Line 46: TSB not defined in full before its use here.

Line 65: "as" missing between "such" and "complement".

Line 294: "overestimation of bactericidal by the OD-based BKA warrants a mechanistic understanding". In this sentence, shouldn't the word "activity" appear between bactericidal and by?

Line 322: "Quantifies" should read "quantities".

Line 629: "verus" should read "versus".

Reviewer's Responses to Questions

Experimental quality

Does each figure have the proper controls?

If 'No', please indicate reasons in Comments for Author box below.

Reviewer #1:

- No

Reviewer #2:

- Yes

Were the data analyzed using appropriate statistical tests?

If 'No', please indicate reasons in Comments for Author box below.

Reviewer #1:

- Yes

Reviewer #2:

- Yes

Reproducibility

Were experiments performed using adequate number of biological replicates?

If 'No', please indicate reasons in Comments for Author box below.

Reviewer #1:

- Yes

Reviewer #2:

- Yes

Does the methods section provide sufficient detail to permit reproducibility?

If 'No', please indicate reasons in Comments for Author box below.

Reviewer #1:

- No

Reviewer #2:

- Yes

Completeness

Are the manuscript's conclusions supported by the data?

If 'No', please indicate reasons in Comments for Author box below.

Reviewer #1:

- No

Reviewer #2:

- Yes

Scholarship

Do the authors cite and discuss the merits of data that would argue for and against their conclusion?

If 'No', please indicate reasons in Comments for Author box below.

Reviewer #1:

- Yes

Reviewer #2:

- Yes

Does the manuscript title & abstract accurately reflect the contents of the manuscript, without hyperbole?

If 'No', please indicate reasons in Comments for Author box below.

Reviewer #1:

- Yes

Reviewer #2:

- Yes

First revision

Author response to reviewers' comments

Thank you for the comments from the editor and the reviewers. We did have (very recent) plasma-based BKA data - because we very much agree with reviewer 1 that we should make evidence-based interpretations. We have included the data (as detailed below) in a succinct format, to not take away from the original aim and scope of the paper. We hope we can add three authors (P. Stephens, C. Davis, and M. Soell) to the paper as they generated and helped analyze the plasma-based BKA data.

Reviewer 1:

1. Major Points

Clarify cell preparation and composition: The manuscript refers to "leukocytes," but the methods describe diluted whole blood without RBC removal. Please clarify whether RBCs were present and, if so, provide cell counts. Discuss how nucleated reptilian RBCs may affect OD readings. If feasible, include a subset experiment with RBC reduced or isolated leukocytes to test OD bias mechanisms.

1. Clarified throughout that our assay made use of blood cells (RBCs and WBCs without plasma). Though we left RBCs in place, calling it whole blood in this context would be incorrect, given that the plasma had been removed.
2. Standardize and report cell numbers and viability: Provide leukocyte counts, viability assessments, and any normalization steps. Without standardization, variability may reflect cell abundance rather than method differences.
 1. We added leukocyte counts in a new Table 3 and a description of differential white blood cell counts in the Methods (lines 168-169).
 2. We addressed known information about cell viability from similar leukocyte-based studies run in our lab (we added citations), but did not have the capacity to include viability assessments in these analyses due to small blood volumes and our preference to conduct bacterial killing assessments as quickly as possible (174-179). Because blood was handled in the same way and only kept on ice very briefly, we do not think we had large differences of viability among samples - and this was also not observed in the studies we cited.
 3. While we realize differences in cell quantities will affect individual variability, every sample was added to both agar plates and 96 well plates. Since we ran paired comparisons on the same sample, this variability should not affect correlation or agreement statistics.
3. Detail bacterial inoculum and calibration: Specify strain and CFU calibration method. Confirm how the 200 CFU inoculum was verified.
 1. Added a section to the methods about preparation and selection of bacterial cultures (lines 185-193). We kept an active culture throughout the sampling seasons. 24 hours prior to sampling, we made and plated serial dilutions up to 1:164000 and maintained dilutions in the fridge until use. Colonies were counted the morning of sampling, and we chose dilutions with concentrations that allowed us to add roughly 5uL of bacterial culture to 145uL of diluted blood.
4. Explain and/or stratify wavelength and temperature analyses: OD readings at 600 nm were limited to fall season. It would be helpful to the reader to understand this experimental decision. Raises the question if analysis should be stratified by temperature.
 1. We added an explanation here (lines 202-204) to explain why data included temperature stratifications. We acknowledged that this was not originally part of our experimental design, but data collected for a project specifically aimed at understanding differences in BKAs based on incubation temperature - for which we happened to try using an absorbance of 600nm, after recognizing that this may give us more congruent results with CFU-based BKAs. All temperatures that we included in the current paper had relatively similar killing - allowing for fair comparisons (i.e., levels of killing were within the optimized conditions). We also cited Painter, 2025 - which includes a description of that experimental design.
5. Correct figure legends and add sample sizes: Bias values in Figure 2 legends do not match text; please correct and annotate N per comparison.
 1. Corrected figure 2 legend to have the correct bias agreements for the second and third Bland-Altman plots (Fig. 2D and 2E). Added N values per comparison.
6. Temper conclusions about plasma based OD BKAs: Current data are cell based; extrapolation to plasma should be framed as a hypothesis or include a brief validation of this claim.
 1. We very much agree that plasma-based interpretations needed data. We actually ran plasma samples in late fall 2025 (roughly the same time as submitting this manuscript) and have included the data in a brief form. We added a short description of sampling to the Introduction (lines 128-130), Methods (lines 166-167; 195-197; 208-209; 216) and ran the same statistics on these paired samples (line 239; 264-265). Results are presented in lines 300-304/324-343 and in a new Fig. 3.

Minor Points

7. Ensure consistent terminology ("cell based BKA" vs "leukocyte based BKA").
 1. We went through the manuscript and ensured consistent wording regarding which components were used.
8. Add interpretation guidance for Lin's concordance coefficient (values near zero = poor agreement; negative = severe discordance).
 1. Added a sentence to the statistical analyses section to give interpretation for Lin's concordance coefficients (lines 258-259).

Reviewer 2:

1. Although not a flaw in the design, I would be interested to know the authors' response to this: Mojave Desert tortoises commonly carry the bacterium *Mycoplasma agassizii* as a subclinical infection. This pathogen is the primary cause of upper respiratory tract disease, and many individuals are lifelong asymptomatic carriers. I am aware the last author has published studies on this pathogen, so is a known expert. However, there is nothing in the methods indicating the subjects were free of this. I feel it is conceivable that subclinical carriers may have different leukocyte profiles or altered innate immune responses compared to uninfected individuals. Could the authors please comment on this and/or alter the methods section to indicate confidence in the infection status of the subjects used in the study?
 1. Clarified in the methods section that animals have been screened for presence of disease, namely *Mycoplasma agassizii* throughout their lives. Molecular testing specifically for *Mycoplasma agassizii* has been performed multiple times on our captive population and has always come up negative. With this, we are confident that the tortoises used in this study are free from infection by *M. agassizii* (lines 161-163).
1. It would be useful for those that will read the paper if the data from the literature review was tabulated with relevant parameters highlighted in specific columns.
 1. We added Table 3 to tally the literature review more clearly.
2. Whilst this isn't a systematic review, I wonder if the authors also searched other databases such as PubMed or Google Scholar as part of the literature review? Web of Science is generally considered more stringent and selective in its journal inclusion criteria than PubMed.
 1. We only searched Web of Science. This was not the most all-encompassing method, but we intended to not include non-ecotoxicological studies and included a statement that clarifies our (admitted) bias (line 145).
3. Something I feel should be added to the discussion: one other element that may be relevant and further validate the central point is interindividual variation in the % of different leukocyte populations in blood. I am not aware of how these parameters vary in tortoises (it may not be known), but if we were to use humans as the reference point, neutrophils show a large amount of variation (in %) between individuals.
 1. We added Table 4 to show mean numbers of leukocyte populations (\pm standard deviation) in our tortoises. Individual variation is quite high in these animals.
4. What was the total number of samples tested? This is not mentioned anywhere and should be.
 1. Added sample sizes for each sampling season to the Animal Care and Sampling section of the methods (lines 169-170). Also added n values for each comparison to figure 2 legend.
5. Normally, I would expect ethical clearance was granted to acquire tortoises and to perform controlled procedures such as taking blood. This information (e.g. a study code and who gave approval) is missing from the methods section and should be added.
 1. Added IACUC information and code to the beginning of the methods section (lines 158-160).

Some brief comments referring to specific parts of the manuscript:

1. Line 116-117: this sentence: "There is a need in the literature to better understand the constitutive function of leukocytes (Sandmeier, 2024a,b)". I agree, however, this is a general point that has widely been discussed, investigated and commented on. I would therefore feel more comfortable if the references (self-citations) for this point were replaced with others from an "external" source.
 1. We replaced the self citations with two other papers from different authors, Ghorai and Priyam, 2018 and Zimmerman, 2020 (line 121).
2. Line 158: "briefly transferred to heparinized collection tubes and stored on ice". Please specify for how long they were on ice. Just to be more accurate and eliminate ambiguity. Of course, sample integrity would be compromised if storage at this temperature

extended beyond 24h.

1. Clarified in the methods how long samples were stored on ice. We typically kept them on ice for a maximum of 30-45 minutes, though this could reach up to an hour on particularly difficult collection days (lines 172-173).
3. Line 163 onwards: A positive control is specified. Would the inclusion of a negative control (blood cells but no bacteria) also be valuable for this?
 1. We do make mention of a negative control on line 171 that was used with the OD-based method as a control against contamination.

Some minor typographical errors I noticed: Line 46: the word "for" is missing before 30. Fixed.

Line 46: TSB not defined in full before its use here. Fixed.

Line 65: "as" missing between "such" and "complement". Fixed.

Line 294: "overestimation of bactericidal by the OD-based BKA warrants a mechanistic understanding". In this sentence, shouldn't the word "activity" appear between bactericidal and by? Fixed.

Line 322: "Quantifies" should read "quantities". Fixed.

Line 629: "verus" should read "versus".

Fixed.

Second decision letter

MS ID#: bio.062411R1

MS Title: 1 Bacterial killing assays in ecoimmunology require cross-validation by agreement statistics

Authors: Mariah Painter; Rylee C. Conklin; Paige Stephens; Courtney Davis; Maxwell Soell; Franziska C. Sandmeier

I am happy to tell you that your manuscript has been accepted for publication in Biology Open, pending our standard publication integrity checks. It was accepted on 22nd March 2026.